# Clinical Outcomes Following Arthroscopic Decompression and Repair versus Repair Alone in Patients with a Concomitant Spinoglenoid Cyst and SLAP Lesion: A Systematic Review

**DOI:** 10.3390/diagnostics13142364

**Published:** 2023-07-13

**Authors:** Du-Han Kim, Hyuk-Joon Sohn, Ji-Hoon Kim, Chul-Hyun Cho

**Affiliations:** Department of Orthopedic Surgery, Keimyung University Dongsan Hospital, Keimyung University School of Medicine, Daegu 42601, Republic of Korea; osmdkdh@gmail.com (D.-H.K.); pandasohn@gmail.com (H.-J.S.); dortn3217@naver.com (J.-H.K.)

**Keywords:** ganglion cyst, SLAP lesion, cyst decompression, arthroscopy, labral repair

## Abstract

(1) Background: Patients with a superior-labrum-from-anterior-to-posterior (SLAP) tear associated with a spinoglenoid ganglion cyst have undergone various procedures. The purpose of this study is to evaluate clinical outcomes following arthroscopic treatment in patients with a concomitant spinoglenoid ganglion cyst and SLAP lesion. (2) Methods: This study followed PRISMA (preferred reporting items for systematic reviews and meta-analyses) guidelines, utilizing the PubMed, EMBASE, Cochrane Library, and Scopus databases. The keywords included shoulder, SLAP, labral tear, spinoglenoid notch, paralabral cyst, arthroscopy, and treatment. (3) Results: A total of 14 articles (206 patients) were included. Repair alone was administered in 114 patients (Group R), and 92 patients underwent additional cyst decompression (Group RD). Both groups showed excellent and similar clinical scores. The rate of the complete resorption of the cyst was 95.5% in Group RD, and 92.2% in Group R. The complication rate was 3.5% in Group RD, and 11.4% in Group R. The reoperation rate was 0% in Group RD, and 5.3% in Group R. (4) Conclusion: Reliable clinical outcomes without serious complications were obtained from the use of both procedures. The decompression of the cyst is a safe method that will alleviate pressure on the suprascapular nerve.

## 1. Introduction

A spinoglenoid ganglion cyst causes vague and nonspecific shoulder symptoms. As a result of advances in diagnostic techniques such as magnetic resonance imaging (MRI), a ganglion cyst around the spinoglenoid notch is no longer a rare disease occurring in the shoulder. The spinoglenoid ganglion cyst is associated with repetitive overhead activities, and a one-way valve mechanism related to the superior-labrum-anterior-to-posterior (SLAP) tear caused by repetitive trauma [1,2]. A cyst that extends to the spinoglenoid notch can press on the nerve, leading to atrophy and weakness in the rotator cuff [3]. Non-operative treatments, such as observation, medication, and needle aspiration are usually recommended, but operative treatment should be considered in patients who show clear signs of nerve compression, or if the discomfort and pain become aggravated despite nonoperative management [4].

Although both open and arthroscopic surgery can be performed, patients commonly undergo arthroscopic surgery [1,5,6,7,8]. Recently, due to the advances in arthroscopic technology, various surgical methods have been introduced; however, there is still controversy regarding arthroscopic surgical techniques [9,10]. Surgical methods can be divided into three main types. Some surgeons have advocated for the repair of the labral tear [2,11,12]; others prefer labral debridement and cyst decompression [1,13]. Both procedures are used by the other groups [14]. Among them, SLAP repair with cyst decompression (the intraarticular or subacromial approach) has become a favored procedure, and good results without complications have been reported [4,15]. However, Schroder et al. recently suggested that labral repair without cyst decompression is a safe and effective procedure, with patients reporting excellent satisfaction. They insisted that the use of their approach might prevent the potential risk of compromising the suprascapular nerve during decompression [14].

Few comprehensive studies on surgical technique have been reported. In particular, only one systematic review on the effect of cyst decompression in the performance of SLAP repair has been reported. In the systematic review by Schroeder et al. [16], they concluded that performing cyst decompression provided no benefits. However, opposing opinions about not treating cysts that are directly compressing the nerve have been reported [17].

Accordingly, the aim of this study was to evaluate the clinical outcomes following arthroscopic decompression and repair, versus repair alone, in patients with a concomitant spinoglenoid cyst and SLAP lesion. The secondary purpose of the study was to determine whether cyst decompression is a dangerous procedure in the treatment of SLAP lesions with concomitant ganglion cysts. We hypothesized that SLAP repair both with and without cyst decompression is a safe and effective procedure.

## 2. Materials and Methods

### 2.1. Search Strategy

The database search was conducted according to PRISMA (preferred reporting items for systematic reviews and meta-analyses) guidelines. An extensive search of the literature, using the PubMed, EMBASE, Cochrane Library, and Scopus databases, was conducted on 2 November 2021. Using a Boolean strategy, the following field search terms were used: search (shoulder) AND (SLAP OR superior labral anterior posterior OR labral tear OR spinoglenoid notch OR paralabral cyst) AND (arthroscopy) AND (treatment). A screening of the citations in the included studies was performed, and unpublished articles were also reviewed through physical searches. A cross-checking of the bibliographies of the relevant articles was subsequently performed, to identify articles that were not identified in the search.

### 2.2. Eligibility Criteria

Studies that met the following criteria were included: (1) the article was in English, (2) the full text was available, and (3) it reported a study on the arthroscopic treatment of SLAP lesions with spinoglenoid ganglion cysts. The exclusion criteria were as follows: (1) the article was not in English, (2) the full text was not available, (3) the article was about open surgery, (4) the article was about SLAP lesions without cysts, (5) there was no specific information on the surgical methods, and (6) there was no information on the pre-or postoperative clinical data (Figure 1).

### 2.3. Study Selection

The studies returned from the initial database search were reviewed independently by two reviewers (DHK and HJS). When a decision could not be reached for a particular article, that article was submitted to a third author (CHC) for review and final decision. Throughout the duration of the search, screening of the content of each article and its reference list was performed, to determine whether there was an overlap of patients from other studies.

### 2.4. Quality Assessment and Data Extraction

The level of evidence provided in the articles was collected. The assessment of the methodological quality of the articles included in this study was performed using criteria from the methodological index for nonrandomized studies (MINORS), a tool validated for use in discerning the methodological quality of nonrandomized studies. The highest possible score is 16 for noncomparative studies, and 24 for comparative studies [18]. The MINORS was applied independently by two blinded authors, and a final score was reached by consensus.

A standardized predetermined criterion form was used for the extraction of all study data. The data extracted for the study characteristics included the first author, the year published, the number of groups included in the study, the type and design of the study, and the level of evidence. The data extracted for demographics included the mean age, the gender ratio, and the mean duration of the follow-up. In addition, data regarding the subjective analysis of outcomes were collected: clinical outcomes (the visual analog scale [VAS] pain score, validated outcome measures, and patient-reported outcomes), the surgical method, muscle power, and MRI results. The data from several patients who experienced complications after surgery and reoperation surgery were extracted for the postoperative complications.

### 2.5. Statistical Methods

Endnote (version 9.0; Thomson Reuters, New York, NY, USA, Clarivate Analysis) was used to manage the data, and the analysis was conducted using SPSS (Version 23.0; IBM Co., Armonk, NY, USA). Fisher’s exact test was performed to assess the statistical significance between both groups; a *p* value of <0.05 was considered statistically significant.

## 3. Results

A total of 21 studies were initially selected for a full-text review. Of these studies, 14 studies met the inclusion criteria [2,4,14,15,19,20,21,22,23,24,25,26,27,28], including 206 patients eligible for the review.

Eleven studies were retrospective cohort studies [2,4,14,15,19,21,24,25,27,28], two studies were case reports [20,26], and one study was a prospective cohort [23]. Using the MINORS criteria, the mean score for included studies was 9.0; a summary of the detailed characteristics of the included studies is shown in Table 1.

A total of 206 patients were finally included, of whom 97 were male and 42 were female. The mean age of the entire group was 40.4 years (with the range 31–48.5 years). The mean follow-up period was 27.6 months. A summary of the detailed demographic data of the included studies is shown in Table 2.

The assessment of pre- and postoperative patient-reported outcome scores for surgery was performed in all the included studies. Eight outcome measures were reported in these studies; the VAS score (six studies) [4,15,21,22,23,25], and the Constant score (five studies) [2,20,21,22,23] were the most commonly used scores. The muscle strength of the shoulder joint was reported in four studies [21,24,26,28]. The outcomes reported by all other patients are shown in Table 3.

### 3.1. Subgroup Analysis

Ten studies reported the results of SLAP repair with cyst decompression (Group RD) [2,4,15,19,20,21,22,23,24,27], and five studies reported the outcomes of SLAP repair without cyst decompression (Group R) [14,23,25,26,28]. Among the 206 patients, 92 patients were included in Group RD, and 114 patients were included in Group R.

In Group RD, the VAS score decreased from 5.8 to 1.3, and the external rotation strength ratio (operation shoulder/contralateral shoulder) improved from 56.85% to 82.31%. The Constant shoulder and Rowe scores improved from 64.2 to 92.9, and from 53.5 to 95.1. In Group R, the VAS score decreased from 7.17 preoperatively to 1.14 postoperatively. The Constant shoulder and Rowe scores improved from 58.5 to 96.75, and from 78.57 to 97.05. Complete resolution was obtained for 95.5% of the patients in Group RD, and the complete resolution rate of the cysts was 92.2% in Group R. However, there were no significant differences between the two groups (all *p* > 0.05) (Table 4).

### 3.2. Complications and Reoperation

None of the included studies reported serious complications. Of the eighty-six patients who reported postoperative complications and underwent reoperation in Group RD, three minor complications (soft tissue infection, adhesive capsulitis, and sensory loss) were reported. Of the one hundred and forty four patients in Group R, thirteen minor complications and six reoperations were reported. The minor complications included seven cases of adhesive capsulitis, one case of persistent labral pain, two cases of persistent subacromial pain, and three cases of persistent pain in the biceps area. Among these patients, one patient who experienced persistent labral pain underwent revision labral repair. Three patients who experienced persistent pain from the long head of the biceps underwent a biceps tenodesis. Two patients who experienced acromial pain underwent subacromial decompression (Table 4).

## 4. Discussion

The primary purpose of the current study was to report on the comprehensive clinical outcomes of SLAP repair with or without cyst decompression in patients with a concomitant SLAP lesion and paralabral cyst. Satisfactory clinical and radiological results from the use of these procedures were reported for the 14 studies and 206 patients included. The authors have also reported excellent results without serious complications.

Among surgical procedures, arthroscopic cyst decompression and SLAP repair are widely performed, and reliable results have been reported [4,21,23]. Bilsel et al. reported satisfactory results from cyst decompression combined with SLAP repair, with or without infraspinatus hypotrophy [21]. They found a type II SLAP in all of the patients during arthroscopy, and debrided the cyst material using a shaver. However, hypotrophy was reported in cases involving a prolonged operation time, and this had a significant effect on the external rotation power and functional outcomes [21]. Shon et al. also performed arthroscopic all-intraarticular decompression and SLAP repair. Twenty patients with paralabral cysts with labral tears underwent surgical procedures. Their decompression technique was conducted as follows. When the torn site of the labrum correlating with a paralabral cyst was identified, the probe was introduced into the cyst toward the spinoglenoid notch, and the cyst was gently but thoroughly decompressed indirectly, by stirring and probing with the blunt probe. The decompression procedure was complete when the cystic fluid was evacuated as much as possible, meaning the fluid would no longer gush out from the hole of the torn labrum. The mean size of the cysts was 2.5 cm. They reported complete cyst removal for 90% of patients, and good-to excellent-clinical results during the mean follow-up period of 42.8 months [15]. Another technique for cyst decompression is the subacromial approach. The location of most spinogleniod cysts is adjunct to the scapular spine. As suggested by Kim et al. [4], using this subacromial approach, identifying the cyst was much easier compared to the intra-articular method, with successful decompression of the cyst without recurrence, as reported by the follow-up magnetic resonance arthrography. Their surgical method was as follows. After the SLAP repair, the ganglion cyst was addressed in the subacromial space. Then, using one or two blunt switching sticks, the interval between the supraspinatus and infraspinatus was gently divided, and dissection was continued, until the ganglion cyst was identified. In addition, direct subacromial decompression can lead to complete decompression in the case of a multilobulated large-sized cyst [29].

SLAP repair alone is another treatment option. Schroder et al., who insisted that a one-way valve mechanism induces the formation of ganglion cysts at the spinoglenoid notch, suggested that closing the valve by repairing the SLAP lesion, without direct decompression, is the most effective method of achieving cyst resolution and relief from symptoms [14]. In their study [14], 37 of 42 patients (88%) who underwent treatment with arthroscopic debridement of the glenoid rim and labral repair showed the complete resolution of the cyst. In the other five patients, the spinoglenoid ganglion cyst was still present, but showed a clear reduction in size. Finally, they insisted that their approach was safe because cyst decompression with labral repair may provide a way to obviate the potential risk of endangering the suprascapular nerve during cyst decompression. Kim et al. [23] conducted a prospective comparison of SLAP repair alone, and SLAP repair with cyst decompression. They grouped patients solely by order after the chosen operation treatment. Group I included 14 patients who received SLAP repair alone, and Group II included 14 patients who received concomitant SLAP repair and cyst decompression. According to their results, no significant difference in the clinical and image results was observed between the two groups. Therefore, they concluded that the use of direct cyst decompression did not result in improved outcomes.

Based on various studies, Schroeder et al. [16] conducted a systematic review of studies, in order to evaluate the clinical outcomes of SLAP repair with or without cyst decompression, and to determine whether cyst management is essential. Sixty-six patients underwent treatment with SLAP repair alone, and ninety-four patients underwent SLAP repair with cyst decompression or excision. According to their findings, the clinical scores (the VAS, Rowe, and Constant scores), the rate of return to work or sports, and the complete resolution of the cyst were excellent, and similar in both groups. Therefore, they concluded that the results did not indicate that cyst decompression provided any advantages [16]. They also insisted that decompression might add more time to the procedure, and could potentially increase the risk of iatrogenic suprascapular nerve injury. Their study included nineteen studies, five more than our study. However, all their included studies were published before 2015, with a relatively short-term follow-up (less than three months). There is also a large difference in the number of papers between the two groups. There are only three studies on SLAP repair alone, while there are seventeen studies on SLAP repair with decompression. Among the three studies on SLAP repair alone, the number of people included in the study by Schroder’s group was 42 out of 66, which was a large number [14]. In addition, some papers that did not present preoperative or postoperative clinical scores were also included. Conversely, our study only included papers that reported clinical scores that are used widely in the shoulder field.

Furthermore, the surgical technique for SLAP repair without cyst decompression is somewhat vague. Damage to the cyst or cystic wall could occur during the labral preparation, or suturing during the performance of SLAP repair. Even though the windowing or decompressing of the cyst by a surgeon is not intentional, the opening of the cyst can occur. In the study reported by Youm et al., which included the SLAP-repair-alone group [28], gelatinous material was found during the preparation of the bone bed in eight out of ten patients. This finding indicated that partial decompression had been achieved by the communicating pathway between the glenohumeral joint and the cyst. They also stated that the cyst was decompressed sublabrally, through the superior labral lesion with blunt instruments and a mechanical shaver during the operation. Nevertheless, their study was included in the SLAP-repair-alone group in Schroeder’s review article [16]. As a result, a special commentary regarding the systematic review reported by the Schroeder group was presented by Burks [17]. According to his commentary, the numbers and data included in the systematic review by Schroeder were insufficient to determine a conclusive result. He also stated that windowing or opening the medial dissection in order to view the cyst and perform direct decompression would be unnecessary. Ultimately, he insisted that a re-evaluation was necessary, to establish some indication for at least performing the decompression of the cyst. He also stated that he would prefer to leave the operating room with some indication that the pressure on the suprascapular nerve had been dealt with, by at least decompressing the cyst. In agreement with Burks [17], we believe that decompressing the cyst using a blunt instrument or arthroscopic shaver is a safe and effective technique. No nerve-related complications were found in the reviewed articles.

No serious complications, such as nerve damage, anchor failure, or infection were reported for either Group R or Group RD. However, SLAP-repair-related complications (adhesive capsulitis, and persistent pain from the long head of the biceps or labral lesion) were reported [14]. Of sixteen patients with complications, four patients underwent additional surgery due to the development of labral-repair-related complications. The optimal treatment of SLAP lesion is debated in the literature, with some authors advocating for direct SLAP repair, and others supporting biceps tenotomy or tenodesis. If the patient is relatively older in age, or shows degenerative changes in the superior labrum, biceps tenodesis could also be a reliable option for the management of SLAP. Biceps tenodesis provides the advantage of allowing for early rehabilitation, which can prevent shoulder stiffness after surgery [3]. Perry et al. performed mini-open combined biceps subpectoral tenodesis and arthroscopic cyst decompression in active-duty military patients. Their included patients presented with chronic shoulder pain and a decreased external rotation strength. The pre- and postoperative assessments showed that the external rotation strength increased from a median of 4 to 5, the ASES score increased from 46.0 to 66.5, and the VAS pain score decreased from a median of 3 to 0. They also reported that all patients returned to full duty, and were able to fulfill all their job requirements without complications, at a mean follow-up of 66 weeks [3].

There are several limitations to our systematic review. Firstly, there are fewer studies reporting on the arthroscopic treatment of labral tears without cyst decompression, than there are studies reporting the results after labral repair with cyst decompression. Although the number of patients was not significant difference between 92 (Group R) and 114 (Group RD), the number of studies showed a two-fold difference. Secondly, most of the articles included in this study reported a low level of evidence. Only one study compared Group RD to Group R, and two studies were case reports. Thirdly, because few studies included the performance of regular follow-up image studies, evaluating the recurrence of cysts was difficult. We also did not perform meta-analyses. The conducting of prospective randomized controlled trials will be required in order to compare the clinical and radiographic outcomes of using SLAP repair alone, and SLAP repair with cyst decompression.

## 5. Conclusions

Reliable clinical outcomes without serious complications were obtained for patients in both Group R and Group RD. The complete removal of the cyst could be dangerous, but minimally decompressing the cyst is a safe method, and will alleviate the pressure on the suprascapular nerve. If it is confirmed that the cyst is compressing the nerve, the establishment of some indication for decompression may be required.

## Figures and Tables

**Figure 1 diagnostics-13-02364-f001:**
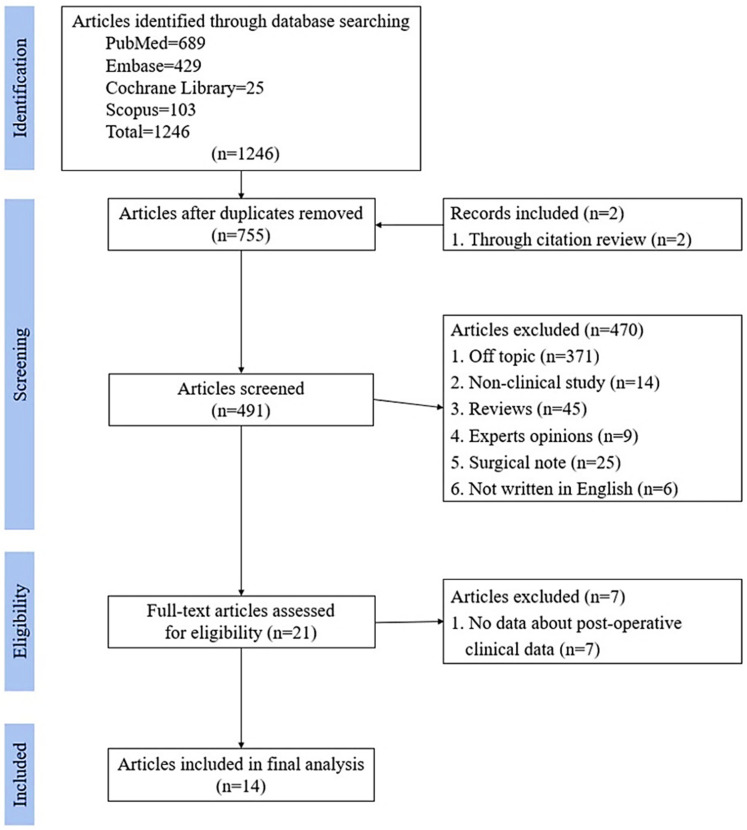
Our PRISMA (preferred reporting items for systematic reviews and meta-analyses) flow diagram.

**Table 1 diagnostics-13-02364-t001:** Study characteristics of the fourteen included studies.

Authors	Year	Number of Shoulders	Type of Study	Design	Level of Evidence	MINORS Score
Lichtenberg S [2]	2004	6	Case series	Retrospective	IV	7/16
Abboud JA [19]	2006	18	Case series	Retrospective	IV	9/16
Baums MH [20]	2006	1	Case report	Retrospective	V	6/16
Westerheide KJ [27]	2006	14	Case series	Retrospective	IV	8/16
Youm T [28]	2006	10	Case series	Retrospective	IV	7/16
Schroder CP [14]	2008	42	Case series	Retrospective	IV	10/16
Pillai G [24]	2011	6	Case series	Retrospective	IV	9/16
Kim DS [23]	2012	28	Comparative study	Prospective	II	21/24
Tan BY [26]	2012	1	Case report	Retrospective	V	6/16
Bilsel K [21]	2014	16	Case report	Retrospective	IV	9/16
Shon MS [15]	2015	22	Case series	Retrospective	IV	9/16
Hashiguchi H [22]	2016	6	Case series	Retrospective	IV	8/16
Kim SJ [4]	2017	26	Case series	Retrospective	IV	9/16
Schrøder CP [25]	2018	47	Case series	Retrospective	IV	8/16

MINOR, methodological index for nonrandomized studies.

**Table 2 diagnostics-13-02364-t002:** The patient demographics.

	Data
No. of patients in study	206
No. of shoulders in study	206
Mean age (years)	40.4 ± 5.1
Gender	
Male	97
Female	42
Mean duration of follow-up (months)	27.6 ± 16.1

**Table 3 diagnostics-13-02364-t003:** The characteristics and clinical outcomes of the included studies (*n* = 14).

Authors	Surgical Method	No. of Patients	Mean Age	Mean Follow-Up (Months)	Preoperative Clinical Outcomes	Last Follow-Up Clinical Outcomes	Return to Work	Return to Sports
Lichtenberg S [2]	Group RD	5	36.6 (24–51)	22.6 (13–40)	Constant 68.4 SST 7.6	Constant 91.8 SST 10.6	NA	NA
Abbound JA [19]	Group RD	8	33 (17–49)	40 (24–58)	ASES 64 Penn shoulder score 74	ASES 95 Penn shoulder score 96	NA	NA
Baums MH [20]	Group RD	1	31	29	Constant 72 EMG positive 1/1	Constant 94 EMG positive 0/1	NA	NA
Westerheide KJ [27]	Group RD	7	41 (27–63)	51 (24–73)	SST 4.3	Constant 94 SST 11.5	NA	NA
Youm T [28]	Group R	10	47.7 (35–56)	10.2 (6–27)	ER weakness 6/10 NC test positive 4/10	ER weakness 0/10 NC test positive 0/10	10/10	10/10
Schroder CP [14]	Group R	42	43 (23–68)	43 (14–108)	Rowe 61.5	Rowe 98.0	39/42	NA
Pillai G [24]	Group RD	6	42 (33–61)	15.2 (12–27)	ER strength ratio 45%	ER strength ratio 85%	NA	NA
Kim DS [23]	Group R Group RD	28	NA	31.4 (24–46) 33.6 (24–58)	Group R VAS 5.4 Constant 58.5 Rowe 52.1 Group RD VAS 5.6 Constant 61.2 Rowe 53.5	Group R VAS 0.6 Constant 96.8 Rowe 94.2 Group RD VAS 0.5 Constant 97.8 Rowe 95.1	Group R 13/14 Group RD 13/14	Group R 13/14 Group RD 13/14
Tan BY [26]	Group R	1	39	22	Weakness of rotator cuff with ISP atrophy	Full recovery of strength Constant 96	1/1	1/1
Bilsel K [21]	Group RD	16	40.5 (32–52)	26 (12–48)	ER strength 61.3 Constant 66.4 VAS 7	ER strength 81.3 Constant 87.3 VAS 2	NA	NA
Shon MS [15]	Group RD	20	38.1 (17–66)	43 (24–77)	VAS 4.5 ± 2.4 ASES 57.9 ± 14.6 SST 7.5 ± 3.6	VAS 1.2 ± 1.3 ASES 90.7 ± 8.6 SST 10.6 ± 1.5	NA	NA
Hashiguchi H [22]	Group RD	6	48.5 (34–70)	63.7 (34–92)	Constant 60.5 VAS 7.7	Constant 97.2 VAS 1.5	NA	NA
Kim SJ [4]	Group RD	9	37.6 (25–61)	24	VAS 3.5 UCLA 21.6 ASES 64.3 SSV 62.9	VAS 0.7 UCLA 32.9 ASES 94.4 SSV 93.1	NA	NA
Schrøder CP [25]	Group R	47	37.0 (17–54)	6 (3–60)	VAS 7.7	VAS 1.3	NA	NA

SST, simple shoulder test; ER, external rotation; VAS, visual analog scale; SSV, subjective shoulder value; UCLA, University of California, Los Angeles shoulder score; ASES, American Shoulder and Elbow Surgeon score; EMG, electromyography; NC, nerve conduction; NA, not available; Group R, repair alone; Group RD, repair and decompression.

**Table 4 diagnostics-13-02364-t004:** The postoperative clinical outcomes and complications in both groups.

	Group RD (*n* = 92)	Group R (*n* = 114)	*p*-Value
Age (years)	39.2	40.6	
Mean follow-up (months)	35.5	23.3	
Clinical outcomes (final)			
VAS	1.3 (*n* = 56)	1.1 (*n* = 61)	
Constant	92.9 (*n* = 42)	96.8 (*n* = 15)	
Rowe	95.1 (*n* = 14)	97.1 (*n* = 56)	
MRI finding			
Complete resorption	95.5% (42/44)	92.2% (94/102)	0.724
Partial resorption	4.5% (2/44)	7.8% (8/102)	0.724
Complications			
Postoperative	3.5% (3/86)	11.4% (13/114)	0.063
Re-operation	0% (0/86)	5.3% (6/114)	0.038 *

VAS, visual analog scale; MRI, magnetic resonance imaging. * Statistically significant. The Group DR complications (*n* = 3) included superficial infection (*n* = 1), adhesive capsulitis (*n* = 1), and skin sensory loss (*n* = 1). The Group R complications (*n* = 13) included adhesive capsulitis (*n* = 7), persistent labral pain (*n* = 1), persistent subacromial pain (*n* = 2), and persistent pain d/t biceps (*n* = 3). The cause of reoperation (*n* = 6) included: persistent labral pain (*n* = 1) = revision repair, persistent subacromial pain (*n* = 2) = SAD, persistent pain d/t biceps (*n* = 3) = biceps tenodesis.

## Data Availability

Not applicable.

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
