# Peer review of "Clinical Outcomes Following Arthroscopic Decompression and Repair versus Repair Alone in Patients with a Concomitant Spinoglenoid Cyst and SLAP Lesion: A Systematic Review"

_diagnostics, 2023, doi:10.3390/diagnostics13142364_

Round 1

Reviewer 1 Report

Introduction should be improved by providing more background information about different procedures for SLAP.

For table 4, it would be appropriate to analysis with Meta analysis.

More references relevant to this research should be cited.

Author Response

Introduction should be improved by providing more background information about different procedures for SLAP.

=> Thank you for your comment. As your comment, we additionally described surgical techniques in the Introduction section. (Line 40-46)

For table 4, it would be appropriate to analysis with Meta analysis.

=> Thank you for your comment. Except for VAS pain score, the difference in the patient of clinical outcomes between the two groups was large, so it was not suitable for meta-analysis. (Constant score 42:15, Rowe score 14:56). We added this in the limitation section.

More references relevant to this research should be cited.

=> Thank you for your comment. We added several references relevant to our study.

Reviewer 2 Report

1. Title would be better if written in PICOS format. At least main outcome of interest should be mentioned. 

for eg. "Clinical outcomes following arthroscopic decompression and repair versus repair alone in patients with a concomitant spinoglenoid cyst and SLAP lesion: A systematic review"

2. looking at the result and conclusion the objective should be:

"to evaluate the clinical outcomes following arthroscopic decompression and repair versus repair alone in patients with a concomitant spinoglenoid cyst and SLAP lesions"

3. Please explain in detail how this study provides new information compared to Schroeder et al Systematic review. both in introduction and discussion section.

4. please provide PROSPERO registration number.

Author Response

  1. Title would be better if written in PICOS format. At least main outcome of interest should be mentioned. 

for eg. "Clinical outcomes following arthroscopic decompression and repair versus repair alone in patients with a concomitant spinoglenoid cyst and SLAP lesion: A systematic review"

=>Thank you for comment. We modified the title.

  1. looking at the result and conclusion the objective should be:

"to evaluate the clinical outcomes following arthroscopic decompression and repair versus repair alone in patients with a concomitant spinoglenoid cyst and SLAP lesions"

=>Thank you for comment. We modified the purpose of study in introduction.

  1. Please explain in detail how this study provides new information compared to Schroeder et al Systematic review. both in introduction and discussion section.

=>Thank you for comment. As your comment, comparing to Schroeder’s review is a key point of our study. So, we explained in detail the difference in Schroeder’s study. Please see line 232-240. Also, we additionally explained the limitation of Schroeder’s review article. (Line 243-251)

  1. please provide PROSPERO registration number.

=>This study was retrospective systematic reviews. Therefore, we don’t have PROSPERO number.

Reviewer 3 Report

Thank you for the opportunity to review this interesting paper. The topic is well-presented, and the analysis is well-conducted. Inclusion and exclusion criteria are well-presented. The literature search flow diagram (Figure 1) is detailed.

Line 14: Please change ‘meta-analyses’ to ‘systematic reviews and meta-analyses’.

 Lines 19-20: There seems to be a clinically significant difference in the complication rate between the two procedures, however, this does not appear to be statistically significant. Please comment. Also, the re-operation rate is statistically significant-please comment also.

Line 27: Please change ‘symptom’ to ‘symptoms’

Line 85: There is no point in blinding reviewers against the authors’ participation in the review. All names of authors are already under the title of the paper!

Line 86: Please add the initials of the third reviewer.

Line 136: Please provide units for strength measurements.

Study 19 by Bilsel et al. is presented as a ‘Comparative study’ in Table 1 when it is not.

Table 4: The mean follow-up times are very different between groups. Please comment.

Limitations of the study are appropriately addressed.

Author Response

Thank you for the opportunity to review this interesting paper. The topic is well-presented, and the analysis is well-conducted. Inclusion and exclusion criteria are well-presented. The literature search flow diagram (Figure 1) is detailed.

Line 14: Please change ‘meta-analyses’ to ‘systematic reviews and meta-analyses’.

->Thank you for comment. We modified that phrase.

 Lines 19-20: There seems to be a clinically significant difference in the complication rate between the two procedures, however, this does not appear to be statistically significant. Please comment. Also, the re-operation rate is statistically significant-please comment also.

->Thank you for comment. We also expected a statistically significant difference in postoperative complications. However, the result was 0.06, close to 0.05. We thought that these results were due to the difference in the number of patients between the two groups (92 vs 114). We described this difference in the limitation section.

Line 27: Please change ‘symptom’ to ‘symptoms.

->Thank you for comment. We modified that word.

Line 85: There is no point in blinding reviewers against the authors’ participation in the review. All names of authors are already under the title of the paper!

-> Thank you for comment. We added the initials of authors.

Line 86: Please add the initials of the third reviewer.

->Thank you for comment. We added the initials.

Line 136: Please provide units for strength measurements.

->Thank you for comment. We added the unit for strength measurements.

Study 19 by Bilsel et al. is presented as a ‘Comparative study’ in Table 1 when it is not.

->Thank you for comment. Bilsel et al. conducted a retrospective comparative study between ISP hypotrophy group and intact ISP group. But, as your comment, their study was classified as case series in our review article. So, we modified it.

Table 4: The mean follow-up times are very different between groups. Please comment.

->Thank you for comment. Schroder CP’s study occupied a large part in repair alone group. Therefore, we think that the follow-up period of a specific group had an impact.

Limitations of the study are appropriately addressed.